# Dietary Liberalization in Tetrahydrobiopterin-Treated PKU Patients: Does It Improve Outcomes?

**DOI:** 10.3390/nu14183874

**Published:** 2022-09-19

**Authors:** Roeland A. F. Evers, Annemiek M. J. van Wegberg, Anita MacDonald, Stephan C. J. Huijbregts, Vincenzo Leuzzi, Francjan J. van Spronsen

**Affiliations:** 1Division of Metabolic Diseases, Beatrix Children’s Hospital, University Medical Center Groningen, 9700 RB Groningen, The Netherlands; 2Dietetic Department, Birmingham Children’s Hospital, Birmingham B4 6NH, UK; 3Department of Clinical Child and Adolescent Studies-Neurodevelopmental Disorders, Faculty of Social Sciences, Leiden University, 2300 RB Leiden, The Netherlands; 4Unit of Child Neurology and Psychiatry, Department of Human Neuroscience, Sapienza University of Rome, 00185 Rome, Italy

**Keywords:** phenylketonuria, tetrahydrobiopterin, diet, systematic review, meta-analysis, nutrition, biomarkers, growth, BMI, weight, mental health, psychosocial functioning, quality of life

## Abstract

Purpose: this systematic review aimed to assess the effects of dietary liberalization following tetrahydrobiopterin (BH_4_) treatment on anthropometric measurements, nutritional biomarkers, quality of life, bone density, mental health and psychosocial functioning, and burden of care in PKU patients. Methods: the PubMed, Cochrane, and Embase databases were searched on 7 April 2022. We included studies that reported on the aforementioned domains before and after dietary liberalization as a result of BH_4_ treatment in PKU patients. Exclusion criteria were: studies written in a language other than English; studies that only included data of a BH_4_ loading test; insufficient data for the parameters of interest; and wrong publication type. Both within-subject and between-subject analyses were assessed, and meta-analyses were performed if possible. Results: twelve studies containing 14 cohorts and 228 patients were included. Single studies reported few significant differences. Two out of fifteen primary meta-analyses were significant; BMI was higher in BH_4_-treated patients versus controls (*p* = 0.02; standardized mean difference (SMD) (95% confidence interval (CI)) = −0.37 (−0.67, −0.06)), and blood cholesterol concentrations increased after starting BH_4_ treatment (*p* = 0.01; SMD (CI) = −0.70 (−1.26, −0.15)). Conclusion: there is no clear evidence that dietary liberalization after BH_4_ treatment has a positive effect on anthropometric measurements, nutritional biomarkers, or quality of life. No studies could be included for bone density, mental health and psychosocial functioning, and burden of care.

## 1. Introduction

Phenylketonuria (PKU; OMIM 261600) is an inborn error of phenylalanine (Phe) metabolism, caused by a deficiency of the Phe hydroxylase (PAH; EC 1.14.16.1) enzyme [1]. Due to this deficiency, the conversion of Phe into tyrosine is impaired, resulting in high Phe concentrations in the blood and brain. These high Phe concentrations ultimately cause severe neurological impairment that typically characterizes the phenotype of untreated PKU. Fortunately, early institution of a life-long dietary treatment that limits intake of Phe by reducing natural protein consumption is very effective in preventing severe complications. Nevertheless, even in early-treated PKU patients, outcomes still appear to be suboptimal, for example, when it comes to cognition and mental health [2], white matter [2], nutrient status [3], bone density [4], overweight and obesity [5], and growth [6].

Tetrahydrobiopterin (BH_4_, prescribed as sapropterin dihydrochloride) is a pharmacological treatment option that aims to further improve outcomes by increasing residual PAH activity in a subset of PKU patients. Through this mechanism, daily administration of BH_4_ in BH_4_-responsive patients may result in lower blood Phe concentrations, and possibly in the increased stability of blood Phe concentrations and/or increased dietary Phe tolerance [7,8,9,10,11]. In BH_4_-responsive patients with suboptimal blood Phe concentrations, BH_4_ can thus be used to improve metabolic control. In already well-controlled BH_4_-responsive patients, however, BH_4_ treatment is used to partially or completely replace dietary treatment, resulting in significant dietary changes [12]. Typically, patients who respond to BH_4_ have a milder phenotype compared to BH_4_-unresponsive patients [13]. Their milder phenotype is caused by a higher level of residual PAH activity, which means that these patients generally require less natural protein restriction, even without BH_4_ treatment.

It has been hypothesised that relaxation of the diet following the start of BH_4_ treatment, accompanied by a reduced need for protein substitutes, may improve many outcomes in PKU patients. It is often said that dietary liberalization may improve quality of life in patients, by decreasing the burden of the strict diet [14,15]. Furthermore, it has been hypothesized that dietary changes with BH_4_ treatment improve growth [16] and nutrition [17]. Possible other clinical benefits of dietary relaxation relate to mental health, psychosocial functioning, burden of care (i.e., the burden experienced by parents and caregivers), and bone health. However, even though BH_4_ has currently been available for more than ten years in many countries, the effect of dietary relaxation associated with BH_4_ treatment on these outcomes requires examination. Therefore, we performed a systematic review and meta-analyses to assess the effects of dietary liberalization following BH_4_ treatment on anthropometric measurements, nutritional biomarkers, quality of life, bone density, mental health and psychosocial functioning, and burden of care in patients with PKU.

## 2. Materials and Methods

### 2.1. Use of Guidelines

This manuscript was written using the Preferred Reporting Items for Systematic Reviews and Meta-Analysis (PRISMA) guidelines [18]. Compliance with these guidelines is described in Appendix A. The review was not registered and the study protocol was not published. 

### 2.2. Search Strategy

The PubMed, Cochrane, and Embase databases were searched up to 7 April 2022. Both medical subject headings and text words were used. The search strategy was generated together with a librarian. The full search criteria are available in Appendix A. 

### 2.3. Eligibility Criteria

The online tool Rayyan^®^ was used to manage the screening process [19]. After deleting duplicate references, studies were screened to select eligible studies for the systematic review. To be eligible for inclusion, studies needed to report on: (1) patients with PKU; (2) treatment with BH_4_ in responsive patients, resulting in dietary liberalization (as measured by an increase in dietary Phe or natural protein intake); and (3) anthropometric measurements, nutritional biomarkers, quality of life, bone density, mental health and psychosocial functioning, and/or burden of care, both before and after starting BH_4_ treatment. Studies were excluded based on the following exclusion criteria: (1) written in a language other than English; (2) containing data only on a short-term BH_4_ loading test for determining (potential) BH_4_ responsiveness; (3) insufficient data for outcome parameters (lacking means with standard deviations (SDs) and lacking any statistical between-subject or within-subject analyses); and (4) case reports, case series, and/or conference abstracts. 

### 2.4. Study Selection

Study selection for the systematic review was conducted in two steps. First, titles and/or abstracts were screened. Second, possible eligible studies were screened by reading the full text. Each step was independently performed by two reviewers (R.A.F.E. and A.M.J.v.W.). Agreement was compared after each step, and disagreements were resolved through discussion until full consensus was reached. After inclusion in the systematic review, studies could be included in one or more meta-analyses if they reported sufficient data on the outcome parameters. 

### 2.5. Data Collection and Data Items

Following the study selection, we collected all data that related to the outcome domains of interest. In addition to outcome data, we collected study and cohort characteristics (study country, study design, type of control group, number of patients, gender, age, length of follow-up, change in blood Phe concentrations, change in dietary Phe intake, change in protein substitute intake). Data from the included studies were collected from their main manuscripts, Appendix A, and online trial data registers (e.g., clinicaltrials.gov, accessed on 11 August 2021). Study authors were contacted for additional information when necessary. Data extraction for meta-analyses was performed by two independent researchers (R.A.F.E. and A.M.J.v.W.) using data collection forms.

### 2.6. Data Analysis

With respect to our objective, two types of analyses were relevant: (1) within-subject analyses, i.e., comparisons between the outcome at baseline (before the start of BH_4_ treatment) and the outcome after a certain follow-up period of BH_4_ treatment; and (2) between-subject analyses, i.e., comparisons between the outcome in BH_4_-treated patients versus the outcome in a control group (e.g., PKU patients not treated with BH_4_), after a certain follow-up period. These analyses are discussed separately in this manuscript.

Meta-analyses were performed if at least two studies or cohorts had similar outcome parameters and reported on those parameters in a manner suitable for a meta-analysis (i.e., reporting a mean, standard deviation, and number of patients/controls). A random-effects model was used for all meta-analyses. For anthropometric outcomes specifically, only data presented in the form of age-corrected z-scores (or SD-scores) were used. All meta-analyses were performed using RevMan 5.

For our primary meta-analyses (Table 1), we used the outcome data from the last point of follow-up, since this study was focused on longer-term effects. In addition, we performed secondary meta-analyses using only short-term data (≤6 months of BH_4_ treatment, using the closest data point up to 6 months) and using only long-term data (≥5 years of BH_4_ treatment, using the last data point). All meta-analyses were additionally stratified for different age groups: <12 years old, 12 to 18 years old, and ≥18 years old. The results of the secondary meta-analyses and stratifications according to age are shown in the Appendix A; when these analyses showed additional significant results (compared to the primary meta-analysis with all age groups combined), it is reported in this manuscript. Furthermore, forest plots of all primary meta-analyses are displayed in Appendix A.

### 2.7. Risk of Bias and Certainty of Evidence

The risk of bias in individual studies was assessed using the Quality In Prognosis Studies (QUIPS) tool [20], which defines six possible areas of bias: study participation, study attrition, prognostic factor measurement, outcome measurement, study confounding, and statistical analyses and reporting. Risk of bias can be scored as low, moderate, or high for each area. Two researchers (R.A.F.E. and A.M.J.v.W.) independently assessed the risk of bias in each paper for each general outcome type, after which agreement was compared and disagreements were resolved through discussion. 

In addition to evaluating the risk of bias in individual studies, reporting bias was also assessed. This was achieved by evaluating the funnel plots of the primary meta-analyses and considering any signs of bias.

Certainty of evidence was evaluated using the Grading of Recommendations, Assessment, Development and Evaluations (GRADE) framework [21]. Each individual outcome was assessed and scored as having either ‘very low’, ‘low’, ‘moderate’, or ‘high’ certainty of evidence. This evaluation was conducted by two independent researchers (R.A.F.E. and A.M.J.v.W.) in a process similar to that used with the QUIPS tool. 

## 3. Results

### 3.1. Study Selection

After removing duplicate publications, a total of 1276 publications were screened (Figure 1). Full texts of 79 reports were ultimately assessed for eligibility, of which 67 were excluded (Appendix A), including 3 papers with overlapping data sets [22,23,24].

### 3.2. Study Characteristics

Twelve studies were included (Table 2). These studies contained fourteen cohorts, since two studies both described two separate cohorts that were analysed individually. First, the study from Aldámiz-Echevarría et al. described two separate cohorts with respect to follow-up time (two years versus five years) [25]. Second, the extension study from Muntau et al. reported on two cohorts with different treatment strategies relating to a previous randomized controlled trial; this difference manifested itself especially with regard to the change in dietary Phe intake after BH_4_ treatment [24,26].

The included studies contained patients from 12 different countries. Ten of the twelve studies were prospective, and seven studies included a control group that consisted of PKU patients who were not treated with BH_4_. In total, the 14 cohorts included 228 patients. The median number of patients per cohort was 14 (range: 6–36). The median percentage of female patients per cohort was 45 (range: 0–67). Mean or median age (reported in 11 cohorts) was below 12 years in seven cohorts, and 12 to 18 years in four cohorts. The median follow-up time after starting BH_4_ treatment was 24 months (range: 3–62 months). 

Following the start of BH_4_ treatment, blood Phe concentrations (reported in 12 cohorts) decreased in seven cohorts and increased in five, although most changes were not significant. Dietary Phe intake increased in all 14 cohorts, reaching significance in 10 cohorts. The mean change in protein substitute intake decrease in all nine cohorts for which this was reported; in three cohorts, this change was found to be significant.

**Table 2 nutrients-14-03874-t002:** ^1^ Last available moment of follow-up was used. ^2^ Or natural protein intake. ^3^ Body weight not taken into account. ^4^ Body weight taken into account. ^5^ Numbers not reported in manuscript; read from graph/figure. ^6^ Mean follow-up time (different follow-up times for different patients). Underlined results denote significant changes within the group of BH_4_-treated patients; if not underlined, there was either no significant change, or statistical analysis was not performed or reported. N/r, not reported. P, prospective. R, retrospective.

Study (Reference)	Main Study Characteristics			Characteristics of BH_4_-Treated Patients at Baseline				Direct Effects of BH_4_ Treatment		
	Country	Study Design	Type of Control Group	Patient Number	Gender (% Female)	Age, Mean ± SD (Years)	Follow-Up Time (Months)	Change ^1^ in Blood Phe Concentrations	Change ^1^ in Dietary Phe Intake ^2^	Change ^1^ in Protein Substitute Intake
Lambruschini 2005 [27]	Spain	P	None	11	64	5.0 ± 4.2	12	16% increase	4.3-fold increase ^3^	100% decrease
Singh 2010 [28]	USA	P	None	6	0	n/r	24	10% decrease ^5^	3.3-fold increase ^4^	84% decrease
Ziesch 2012 [29]	Germany	P	Non-BH_4_-treated PKU patients	8	50	11.1 ± 4.4	3	7% increase	3.4-fold increase ^3^	n/r
Aldámiz-Echevarría 2013 (1) [25]	Spain	R	Non-BH_4_-treated PKU patients	36	50	5.0 ± 4.6	24	43% increase	1.4-fold increase ^4^	44% decrease
Aldámiz-Echevarría 2013 (2) [25]				10	40	5.2 ± 3.1	60	42% increase	1.2-fold increase ^4^	57% decrease
Demirdas 2013 [30]	The Netherlands	P	Non-BH_4_-treated PKU patients	10	n/r	13.8 ± 9.7	n/r	n/r	4.1-fold increase ^3^	n/r
Douglas 2013 [31]	USA	P	Non-BH_4_-treated PKU patients	11	n/r	n/r	12	33% decrease ^5^	3.8-fold increase ^3,5^	85% decrease ^5^
Scala 2015 [32]	Italy	P	None	17	n/r	n/r	62 ^6^	53% increase	1.7-fold increase ^3^	n/r
Tansek 2016 [33]	Slovenia	P	None	9	n/r	6.2 ± 3.1	24	5% decrease	3.2-fold increase ^3^	93% decrease
Feldmann 2017 [34]	Germany	P	Non-BH_4_-treated PKU patients	20	35	12.5	6	n/r	2.6-fold increase ^4^	42% decrease
Brantley 2018 [35]	USA	P	Healthy controls and non-BH_4_-treated PKU patients	18	44	16.6 ± 10.3	12	23% decrease	1.5-fold increase	66% decrease
Evers 2018 [36]	The Netherlands	R	Non-BH_4_-treated PKU patients	21	67	13.1 ± 9.2	60	3% decrease ^5^	1.5-fold increase ^4^	68% decrease
Muntau 2021 (1) [26]	Austria, Belgium, Czech Republic, Germany, Italy, The Netherlands, Slovakia, Turkey, UK	P	None	25	40	1.7 ± 1.0	36	6% decrease ^5^	2.0-fold increase ^5^	n/r
Muntau 2021 (2) [26]				26	46	1.7 ± 1.0	36	12% decrease ^5^	1.1-fold increase ^5^	n/r

### 3.3. Results of the Systematic Review and Meta-Analyses

#### 3.3.1. Anthropometric Measurements

##### Results of Individual Studies

Anthropometric measurements were investigated in nine cohorts. In total, 5 out of 41 analyses revealed significant results. Weight [36], BMI [32], height [28], and brachial adipose area [27] were found to have significantly increased in single cohorts (Table 3). In contrast, in one cohort, weight was lower in BH_4_-treated patients after the follow-up than in non-BH_4_-treated PKU patients [25].

##### Results from Meta-Analyses

One out of nine of the primary meta-analyses performed showed a significant result: BMI was significantly higher following BH_4_ treatment compared to non-BH_4_-treated PKU patients (Table 1). Furthermore, a secondary meta-analysis for short-term between-subject differences showed a significant (*p* = 0.01) result that indicated higher weight in BH_4_-treated patients, whereas the primary and long-term meta-analyses for this parameter revealed no significant results.

#### 3.3.2. Nutritional Biomarkers

##### Results of Individual Studies

Nutritional biomarkers (not including blood Phe concentrations) were assessed in five cohorts. Out of 43 analyses, 10 were significant (Table 4). Significant increases were found in cholesterol [36], transthyretin [28], selenium [27], and haematocrit [28] (all single cohorts), and haemoglobin [28,36] (two cohorts). One cohort showed decreases in methylmalonic acid [36] (indicating better intracellular vitamin B12 status) and phosphate [36]. Brantley et al. reported significant differences for vitamin B12, but only for BH_4_-treated patients <18 years, in whom vitamin B12 concentrations significantly dropped after starting BH_4_ treatment and were also lower when compared to non-BH_4_-treated PKU patients [35]. 

##### Results from Meta-Analyses

One of the three primary meta-analyses was significant (Table 1). This analysis indicated an increase in cholesterol in BH_4_-treated patients. 

#### 3.3.3. Quality of Life

##### Results of Individual Studies

Quality of life was investigated in four cohorts. Studies investigating quality of life using generic questionnaires found no significant differences between total scores (Table 5) [29,30,34]. With quality of life subscales, only one significant result was reported in one study: self-esteem (proxy report) was significantly higher in BH_4_-treated patients compared to non-BH_4_-treated PKU patients (*p* = 0.030) [29].

Demirdas et al. additionally measured quality of life with questionnaire for patients with a chronic illness, but this did not demonstrate a significant within-subject or between-subject difference in quality of life [30]. Only Douglas et al. used a PKU-specific questionnaire, and reported an increase in quality of life after 12 months of BH_4_ treatment [31]. Furthermore, this study also reported significant within-subject improvement in the subscales ‘impact’ and ‘satisfaction’ [31]. These improvements were significantly associated with increased Phe tolerance. 

One study assessed parental quality of life (Table 5) [34]. This study performed a between-subject analysis but found no difference in the total scores, although it was reported that the subscale ‘emotional stability’ was significantly higher in the BH_4_ group (*p* = 0.037). 

##### Results from Meta-Analyses

Meta-analyses could only be performed with data from studies that assessed quality of life through generic questionnaires. None of the four meta-analyses showed significant results (Table 1). 

#### 3.3.4. Bone Density, Mental Health and Psychosocial Functioning, and Burden of Care

No studies could be included for bone density, mental health and psychosocial functioning, and burden of care.

### 3.4. Assessment of Risk of Bias Assessment and Certainty of Evidence

#### 3.4.1. Risk of Bias in Individual Studies

Risk of bias was low or moderate for most areas in most studies (Appendix A). Of the 108 total ratings, 53 were low (49.1%), 54 were moderate (50.0%), and one was high (0.9%). Moderate risks of bias often existed for the domains ‘study participation’ and ‘study confounding’, whereas bias ‘statistical analysis and reporting’ was usually low.

#### 3.4.2. Risk of Reporting Bias

The funnel plots of the primary meta-analyses are given in Appendix A. Although interpretation is hindered due to the low number of publications for most topics, we did not find clear signs of reporting bias.

#### 3.4.3. Certainty of Evidence

Certainty of evidence, assessed using the GRADE method, was “low” or “very low” for all parameters. This was due to the mostly observational study designs, low sample sizes, and inconsistent outcomes for some parameters (Appendix A).

## 4. Discussion

A synthetic form of BH_4_, sapropterin dihydrochloride (Kuvan™), was the first pharmaceutical treatment option approved for PKU. Following its approval by the FDA [37] and EMA [38], and its subsequent recommendations by the American and European guidelines for use in responsive patients [15,39], BH_4_ has become part of standard PKU care. In many BH_4_-responsive patients, BH_4_ treatment results in a higher dietary Phe tolerance. While it has previously been hypothesized that such dietary relaxation has beneficial effects, this systematic review found no clear evidence for improvements in anthropometric measurements, nutritional biomarkers, quality of life, bone density, mental health and psychosocial functioning, and burden of care.

We will first address the strengths and limitations of the methodology we used for this systematic review. Since we aimed to collect evidence on all of the possibly relevant parameters, we assessed all data related to the outcome domains from the included studies. While this clearly resulted in a less focused systematic review, we considered it our only option for giving a broad overview of the evidence on the effects of BH_4_ treatment. In line with these considerations, our inclusion and exclusion criteria were broad (e.g., we did not select studies based on follow-up time or patients’ age), creating a rather heterogenous sample of cohorts. While we have homogenized our results by performing several types of secondary meta-analysis, the method behind our study selection prevents us from coming to more specific or detailed conclusions.

In this systematic review, we assessed outcomes related to six domains: anthropometric measurements, nutritional biomarkers, quality of life, bone density, mental health and psychosocial functioning, and burden of care. Although outcomes are, for the most part, comparable to those of a healthy population, problems have previously been noted in this domains among PKU patients [2,3,4,6,40], thus leaving room for improvement. However, the findings of this systematic review indicate that (1) for many domains of interest, insufficient data exist; and (2) for domains with sufficient data, few signs of improvement following dietary relaxation due to BH_4_ treatment are seen (Table 6). Considering this latter finding, only two primary meta-analyses showed significant results. However, these analyses indicated a higher BMI compared to non-BH_4_-treated PKU patients and an increase in blood cholesterol concentrations following BH_4_ treatment, which are not positive changes.

For anthropometric measurements, it has been hypothesized that an increase in intact protein intake could impact height and growth [16]. However, this was not found in our meta-analysis. As was the case with other outcomes, this could be due to limitations in the data, such as small sample sizes, mixed age groups, and follow-up periods that were too short. However, it may also indicate that dietary liberalization does not improve this outcome, since these measurements were generally already normal prior to starting BH_4_ treatment. For weight and BMI, the meta-analysis found significantly higher BMI but not weight in BH_4_-treated PKU patients compared to non-BH_4_-treated PKU patients. This discrepancy appears to be at least partly driven by the first cohort in the study from Aldamiz et al., in which BH_4_-treated patients had a higher BMI z-score but a lower weight z-score compared to dietary-treated PKU patients, in contrast to the other cohorts that showed higher BMI and weight in BH_4_-treated patients [25]. A secondary meta-analysis for short-term effects showed that weight was significantly higher compared to non-BH_4_-treated PKU patients, but this was not observed in the long-term analysis. With regard to within-subject studies, one study reported an increase in BMI after BH_4_ treatment, but this was not the case in other studies and was hence not replicated in our within-subject meta-analysis. Despite the significant findings for weight and BMI, mean/median weight and mean/median BMI in BH_4_-treated patients typically remained within one standard deviation of age- and gender-adjusted reference data, with the exception of two cohorts (weight z-score of 1.21 [36]; BMI z-score of 1.27 [33]). Although these results are somewhat conflicting, it could be hypothesized that relaxing the diet in BH_4_-treated patients may result in unhealthy food choices in some patients, especially in the short term [41,42], and therefore influences body weight. However, most of the studies included in this systematic review lack the specific dietary data necessary to test this hypothesis. Only Singh et al. reported relevant dietary data: in line with the fact that weight remained stable in their cohort, intake of total calories, fats and proteins had not significantly changed in BH_4_-treated patients [28]. Other studies, not included in this systematic review, similarly showed no increase in intake of total proteins, carbohydrates, fats, or calories [41,42]. Nevertheless, our findings do underline the importance of continued, or possibly even intensified, nutritional counselling and education to ensure a balanced diet following BH_4_-related dietary relaxation. 

Regarding nutritional biomarkers, many were only assessed in single studies and meta-analyses could thus often not be performed. For cholesterol, however, a within-subject meta-analysis could be performed, which indicated increased blood cholesterol concentrations in BH_4_-treated patients. This is somewhat in line with the results we found for weight and BMI, and may result from a less healthy diet following BH_4_ treatment. Furthermore, while several studies did report lower dietary intake, in some cases even the below recommended amounts, of several micronutrients [35,41,42], these dietary findings were generally not mirrored by changes in the biomarker concentrations of these micronutrients. In fact, the significant results found in single studies typically indicated an improvement, although these changes are possibly related to ageing effects (e.g., haemoglobin generally increases in children). 

Quality of life is probably one of the most important outcome measures with regard to dietary relaxation with the use of BH_4_. In contrast to experiences in practice, the meta-analysis did not show any improvement in quality of life following BH_4_ treatment. There are several possible explanations for this. First, this observation might be an example of hedonic adaptation, i.e., the psychological phenomenon describing the relative stability of someone’s happiness, irrespective of both positive and negative life events [43]. Moreover, it may also be explained by a ceiling effect: BH_4_ treatment is most effective in patients with a relatively mild phenotype [13], in whom quality of life is already very good [44]. In addition, many BH_4_-treated patients cannot fully liberalize their diet, meaning that they may still experience the negative consequences of dietary treatment. Lastly, these counterintuitive results may be caused by a methodological problem relating to the sensitivity of quality of life questionnaires. The only study that used a PKU-specific questionnaire actually found an increase in quality of life after 12 months of BH_4_ treatment, which was related to increased Phe tolerance [31]. Since this result is in line with reported patient experiences in clinical practice, it can be concluded that general quality of life questionnaires are not always sensitive enough for PKU patients, as they do not seem to capture the daily burden of a strict diet. While general questionnaires may suffice to detect relatively large differences, the use of a PKU-specific questionnaire is recommended over non-specific questionnaires to be able to find relatively subtle changes in quality of life [45].

For bone density, mental health and psychosocial functioning, and burden of care, we could not include any studies that assessed the effects of BH_4_ treatment in a longitudinal manner. Although some of these domains may partly overlap with quality of life, no conclusions can be drawn for these outcomes. For example, burden of care refers in particular to emotional, financial, and social problems faced by caregivers. Although these problems may in part be covered by parental quality of life, for which we could include one study, the effect of BH_4_ treatment on this domain specifically deserves attention in the future. In line with the scarce data for some other parameters, the complete lack of studies for bone density, mental health and psychosocial functioning, and burden of care underline the need for additional research on the effects of dietary liberalization. 

Although this parameter lay outside of the scope of our study, two of the included studies investigated IQ and reported similar IQ levels after dietary relaxation [26,27]. This was expected, since blood Phe concentrations did not change dramatically after BH_4_ treatment in these patients (382 to 442 µmol/L in Lambruschini et al. [27]; approximately 270 to 255 µmol/L and 330 to 290 µmol/L, respectively, in the two cohorts in Muntau et al. [26]). Relatedly, there are several studies in which BH_4_ treatment was used to decrease blood Phe concentrations rather than adjusting natural protein intake. It is noteworthy that some of these studies reported improved outcomes on executive functioning and neuro-imaging findings when Phe levels decreased with BH_4_ treatment (Appendix A). 

Since there is currently no strong evidence for improved outcomes after BH_4_-induced dietary relaxation, and the costs of BH_4_ treatment are high (data from the United States indicate that average annual costs are approximately $67,000 and $122,000 for children and adults, respectively) there are questions as to whether dietary relaxation justifies the use of BH_4_ in all responsive patients, especially when they still need to use protein substitutes. However, as mentioned, most quality of life reports used non-specific questionnaires that do not seem to accurately capture current practice experiences. Thus, the lack of findings in this systematic review, especially for quality of life but also for the other outcomes, could be explained by a lack of reliable data, which is partly reflected by the ‘low’ and ‘very low’ GRADE ratings. The low number of studies also prevented us from conducting more detailed analyses to control for differences in the increase in natural protein intake. Therefore, our findings do not argue against the use of BH_4_ to increase natural protein intake, but rather underscore the need for additional understanding of the effects of BH_4_-related dietary relaxation, especially on quality of life. PKU-specific quality of life questionnaires, which already are available [44,46], could be used for this purpose. Other recommendations for future research on dietary liberalization in PKU patients are summarized in Table 6. 

## 5. Conclusions

Even though BH_4_ treatment can result in higher dietary Phe tolerance, this systematic review shows that there is a lack of evidence that dietary relaxation leads to improvements in anthropometric measurements, nutritional biomarkers, and quality of life. Furthermore, no studies could be included for bone density, mental health and psychosocial functioning, and burden of care. The results for quality of life are especially surprising, since they are in sharp contrast with observations from clinical practice. Overall, these results necessitate future studies on BH_4_ and other potential drugs that aim to allow dietary liberalization in PKU patients.

## Figures and Tables

**Figure 1 nutrients-14-03874-f001:**
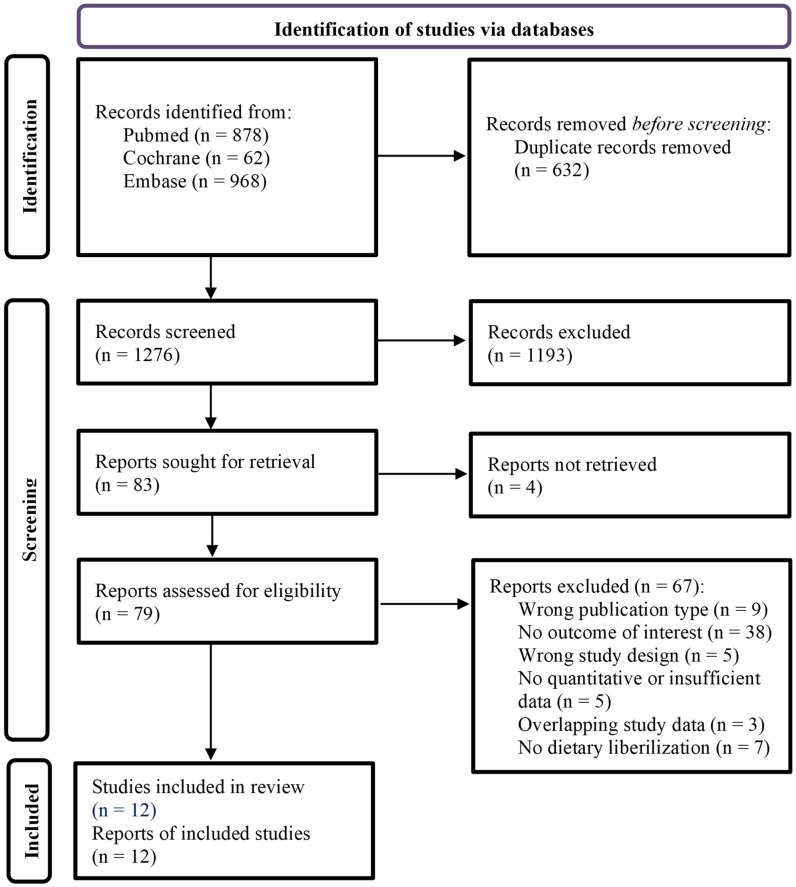
Flowchart of the study inclusion.

**Table 1 nutrients-14-03874-t001:** Overview of the results from the primary meta-analyses. Means for cohort characteristics are weighted by the number of patients per cohort, and are based on full cohort data as presented in Table 2. ↑, higher after BH4 treatment (within-subject analysis) or higher in BH4-treated patients after follow-up compared to a control group (between-subject analysis). N/r, not reported.

Parameter		Main Result	Patient Number	Included Cohorts	Weighted Means Based on Full-Cohort Data (*Means in Individual Cohorts*)			
					Age (Years)	Follow-Up Time (Months)	Change in Blood Phe Concentrations (%)	Change in Dietary Phe Intake (Fold Increase)
**Anthropometric measurements**								
Weight	Within-subject	*p* = 0.61	67	Aldámiz-Echevarría 2013 (1) and (2); Evers 2018	7.6 (*5.0; 5.2; 13.1*)	41 (*24; 60; 60*)	28 (*43; 42; −3*)	1.4 (*1.4; 1.2; 1.5*)
	Between-subject	*p* = 0.95						
BMI	Within-subject	*p* = 0.82	73	Singh 2010; Aldámiz-Echevarría 2013 (1) and (2); Evers 2018	7.6 (*n/r; 5.0; 5.2; 13.1*)	39 (*24; 24; 60; 60*)	25 (*−10; 43; 42; −3*)	1.6 (*3.3; 1.4; 1.2; 1.5*)
	Between-subject	*p* = 0.02, ↑	67	Aldámiz-Echevarría 2013 (1) and (2); Evers 2018	7.6 (*5.0; 5.2; 13.1*)	41 (*24; 60; 60*)	28 (*43; 42; −3*)	1.4 (*1.4; 1.2; 1.5*)
Height	Within-subject	*p* = 0.80	62	Singh 2010; Aldámiz-Echevarría 2013 (1) and (2); Evers 2018	7.6 (*n/r; 5.0; 5.2; 13.1*)	39 (*24; 24; 60; 60*)	25 (*−10; 43; 42; −3*)	1.6 (*3.3; 1.4; 1.2; 1.5*)
	Between-subject	*p* = 0.43	56	Aldámiz-Echevarría 2013 (1) and (2); Evers 2018	7.6 (*5.0; 5.2; 13.1*)	41 (*24; 60; 60*)	28 (*43; 42; −3*)	1.4 (*1.4; 1.2; 1.5*)
Growth velocity	Within-subject	*p* = 0.20	56	Aldámiz-Echevarría 2013 (1) and (2); Evers 2018	7.6 (*5.0; 5.2; 13.1*)	41 (*24; 60; 60*)	28 (*43; 42; −3*)	1.4 (*1.4; 1.2; 1.5*)
	Between-subject	*p* = 0.73						
**Nutritional biomarkers**								
Albumin	Within-subject	*p* = 0.40	38	Lambruschini 2005; Singh 2010; Evers 2018	10.3 (*5.0; n/r; 13.1*)	40 (*12; 24; 60*)	−8 (*−16; −10; −3*)	1.5 (*4.3; 3.3; 1.5*)
Cholesterol	Within-subject	*p* = 0.01, ↑	27	Singh 2010; Evers 2018	13.1 (*n/r; 13.1*)	52 (*24; 60*)	−5 (*−10; −3*)	1.9 (*3.3; 1.5*)
Haemoglobin	Within-subject	*p* = 0.11						
**Quality of life**								
HR-QoL (patient-report)	Within-subject	*p* = 0.86	33	Ziesch 2012; Demirdas 2013; Feldmann 2017	12.5 (*11.1; 13.8; 12.5*)	5 (*3; n/r; 6*)	7 (*7; n/r; n/r*)	3.2 (*3.4; 4.1; 2.6*)
	Between-subject	*p* = 0.92						
HR-QoL (proxy-report)	Within-subject	*p* = 0.14	33	Ziesch 2012; Demirdas 2013; Feldmann 2017	12.5 (*11.1; 13.8; 12.5*)	5 (*3; n/r; 6*)	7 (*7; n/r; n/r*)	3.2 (*3.4; 4.1; 2.6*)
	Between-subject	*p* = 0.22						

**Table 3 nutrients-14-03874-t003:** Results of within-subject (baseline versus last point of follow-up during BH_4_ treatment) and between-subject (BH_4_-treated group versus control group at last point of follow-up) analyses for different parameters regarding anthropometric measurements. Cohort characteristics are taken from Table 2. =, no significant change (within-subject analyses) of no significant difference (between-subject analyses); ↑, significant increase or significantly higher in BH_4_-treated group (if not already significantly higher at baseline); ↓, significant decrease or significantly lower in BH_4_-treated group (if not already significantly lower at baseline). N/r, not reported.

	Results							Cohort Characteristics				
Study/Analyses	Weight	Body Mass Index	Height	Growth	Head Circumference	Brachial Muscular Area	Brachial Adipose Area	Number of Patients	Age at Baseline (Years)	Follow-Up Time (Months)	Change in Blood Phe Concentration	Fold Increase in Dietary Phe Intake
**Within-subject analyses**												
Lambruschini 2005 [27]	=		=			=	↑	11	5.0 ± 4.2	12	+16%	4.3
Singh 2010 [28]		=	↑					6	n/r	n/r	−10%	3.3
Aldámiz-Echevarría 2013 (1) [25]	=	=	=	=				36	5.0 ± 4.6	24	+43%	1.4
Aldámiz-Echevarría 2013 (2) [25]	=	=	=	=				10	5.2 ± 3.1	60	+42%	1.2
Scala 2015 [32]		↑						17	n/r	62	+53%	1.7
Tansek 2016 [33]		=	=					9	6.2 ± 3.1	24	−5%	3.2
Evers 2018 [36]	↑	=	=	=				21	13.1 ± 9.2	60	−3%	1.5
Muntau 2021 (1) [26]	=	=	=		=			25	1.7 ± 1.0	36	−6%	2.0
Muntau 2021 (2) [26]	=	=	=		=			26	1.7 ± 1.0	36	−12%	1.1
**Between-subject analyses**												
Aldámiz-Echevarría 2013 (1) [25]	↓	=	=	=				36	5.0 ± 4.6	24	+43%	1.4
Aldámiz-Echevarría 2013 (2) [25]	=	=	=	=				10	5.2 ± 3.1	60	+42%	1.2
Evers 2018 [36]	=	=	=	=				21	13.1 ± 9.2	60	−3%	1.5

**Table 4 nutrients-14-03874-t004:** Results of within-subject (baseline versus last point of follow-up during BH_4_ treatment) and between-subject (BH_4_-treated group versus control group at last point of follow-up) analyses for different parameters regarding anthropometric measurements. Cohort characteristics are taken from Table 2. =, no significant change (within-subject analyses) of no significant difference (between-subject analyses); ↑, significant increase or significantly higher in BH_4_-treated group; ↓, significant decrease or significantly lower in BH_4_-treated group. * No significant difference for patients > 18 years, but a significant difference for patients < 18 years. N/r, not reported.

	Results																					Cohort Characteristics				
Study/Analyses	Cholesterol	Triglycerides	Iron	Transferrin	Ferritin	Albumin	Total Protein	Transthyretin	Vitamin A	Vitamin B6	Folate	Vitamin B12	MMA	Calcifedol	Vitamin E	Calcium	Phsophate	Selenium	Zinc	Haemoglobin	Haematocrit	Number of Patients	Age at Baseline (Years)	Follow-Up Time (Months)	Change in Blood Phe Concentration	Fold increase in Dietary Phe Intake
**Within-subject analyses**																										
Lambruschini 2005 [27]					=	=			=		=	=			=			↑	=			11	5.0 ± 4.2	12	+16%	4.3
Singh 2010 [28]	=					=	=	↑												↑	↑	6	n/r	n/r	−10%	3.3
Tansek 2016 [33]												=						=	=			9	6.2 ± 3.1	24	−5%	3.2
Brantley 2018 [35]			=							=	=	↓/= *										18	16.6 ± 10.3	12	−23%	1.5
Evers 2018 [36]	↑	=		=		=							↓	=		=	↓			↑		21	13.1 ± 9.2	60	−3%	1.5
**Between-subject analyses**																										
Brantley 2018 [35]			=							=	=	↓/= *										18	16.6 ± 10.3	12	−23%	1.5
Evers 2018 [36]	=	=		=		=							=	=		=	=			=		21	13.1 ± 9.2	60	−3%	1.5

**Table 5 nutrients-14-03874-t005:** Results of within-subject (baseline versus last point of follow-up during BH_4_ treatment) and between-subject (BH_4_-treated group versus control group at last point of follow-up) analyses for total scores of quality of life questionnaires. Cohort characteristics are taken from Table 2. =, no significant change (within-subject analyses) of no significant difference (between-subject analyses); ↑, significant increase or significantly higher in BH_4_-treated group (if not already significantly higher at baseline). N/r, not reported.

	Results					Cohort Characteristics				
Study/Analyses	Generic HR-QoL of Patients		Specified HR-QoL of Patients		Parental QoL	Number of Patients	Age at Baseline (Years)	Follow-Up Time (Months)	Change in Blood Phe Concentration	Fold increase in Dietary Phe Intake
	Patient Report	Proxy Report	Patient Report	Proxy Report						
**Within-subject analyses**										
Ziesch 2012 [29]	=	=				8	11.4 ± 4.4	3	+7%	3.4
Demirdas 2013 [30]	=	=	=	=		10	13.8 ± 9.7	n/r	n/r	4.1
Douglas 2013 [31]			↑			11	n/r	12	−33%	3.8
**Between-subject analyses**										
Ziesch 2012 [29]	=	=				8	11.4 ± 4.4	3	+7%	3.4
Demirdas 2013 [30]	=	=	=	=		10	13.8 ± 9.7	n/r	n/r	4.1
Douglas 2013 [31]			=			11	n/r	12	−33%	3.8
Feldmann 2017 [34]	=	=			=	20	12.5	6	n/r	2.6

**Table 6 nutrients-14-03874-t006:** Overview of our main findings, and recommendations for care and future research on dietary liberalization, resulting from adjuvant treatment options such as BH_4_, in PKU patients. BMI, body mass index; DEXA, dual-energy X-ray ab-sorptiometry; PKU, phenylketonuria; QoL, quality of life.

Domain	Data Available	Main Conclusion	Recommendations for Care	Domain-Specific Recommendations for Research
**Anthropometric measurements**	Data from nine cohorts.	No clear effect on most anthropometric measures; possibly a small increase in weight and BMI.	Physicians and dieticians should pay attention to the nutritional status of BH_4_-treated PKU patients; continued nutritional counselling and education are important to ensure a balanced diet.	Anthropometric measurements should include z-scores for weight, height, and BMI for all patients, and head circumference and growth speed for paediatric patients specifically.
**Nutritional biomarkers**	Data from five cohorts.	No clear effect on most nutritional biomarkers; possibly an increase in blood cholesterol concentrations.		Nutritional biomarkers should at least include homocysteine and/or MMA, haemoglobin, MCV, and ferritin. Other biomarkers, such as cholesterol and 25-hydroxyvitamin D, may be added.
**Quality of life**	Data from four cohorts.	No clear effect on quality of life.		Issues such as QoL, psychosocial outcomes and mental health, and burden of care should be assessed with the use of PKU-specific (instead of generic) questionnaires, such as the PKU-QoL questionnaire.
**Burden of care**	No available data.			
**Mental health and psychosocial functioning**	No available data.			
**Bone density**	No available data.			Bone density measurements should be investigated (e.g., using DEXA).
**General recommendations for research**	- Ensure a sufficiently large cohort size; perform studies in a multicenter setting if necessary to recruit a sufficient number of patients. - Include a control group to facilitate the interpretation of outcomes in the treatment group. Ideally, the control group would consist of PKU patients who are not being treated with BH_4_ and who are similar to the treatment group with respect to age, gender, metabolic control, and dietary treatment (i.e., natural protein intake) at baseline. - Report raw outcome data (means and standard deviations) to allow for data merging in meta-analyses.			

## Data Availability

Not applicable.

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
