# Peer review of "Dietary Liberalization in Tetrahydrobiopterin-Treated PKU Patients: Does It Improve Outcomes?"

_nutrients, 2022, doi:10.3390/nu14183874_

Round 1

Reviewer 1 Report

The authors present a systematic review on the effects of dietary liberalization following BH4 treatment in patients with PKU. The outcome domains anthropometric measurements, nutritional biomarkers, quality of life, bone density, mental health and psychosocial functioning, and burden of care were evaluated. The authors included twelve studies containing 14 cohorts and 228 patients. 

This is an important evaluation as BH4 has become part of the standard PKU treatment resulting in a higher Phe tolerance in responsive patients. In contrast to current hypotheses and clinical observations the authors did not find beneficial effects on anthropometric measures, nutritional biomarkers and quality of life. There were insufficient data for systematic evaluation of the other domains, a weakness of the study, which the authors acknowledge and discuss accordingly. 

The rather surprising result that patients do not seem to benefit from dietary relaxation is sufficiently discussed and the authors present recommendations for improved study design and reporting.  

Minor comments:

Page 5/6: Table 1 is unclear and hard to read. In the column "Study design" the letters "P" and "R" could be used for prospective and retrospective studies leaving more space for the other columns. Footnotes 8 and 9 have been used in the figure legend but are missing in the table. 

Page 7 line 215, page 8 line 243, Page 9 line 272: a reference is made to Table 2, which is missing. The authors most likely mean Table S3.1 in the supplement. However, such an important table should be included in the main manuscript. 

Some minor spelling errors should be corrected (Table 6: "Anthropometric" instead of "Anthropomorphic"; General recommendations for research: "PKU patients" instead of "PKU patient"; )

Reviewer 2 Report

Thankyou for your interesting and relevant study.

There are some minor edits that could improve readability

L51 aims rather than aimed, L68 requires rather than required. L105 if instead of in case, L139 when instead of in case, L351 therefore instead of thereby, L 384 to increased phe tolerance rather than with increased phe tolerance, L429, 'a lack of' rather than 'hardly any'. 

Table 6: lay out could be improved

Comment regarding interpretation of growth in paragraph starting L331: BH4 may not improve growth outcomes, particularly height,  if normal prior to commencement. 

Reviewer 3 Report

The longterm effect of BH4 is a “hot” topic in PKU-science. Therefore a manuscript reviews the studies of BH4 therapy can be of great interest. However, perform a meta-analysis in a rare disease is always challenging. Due to the low number of studies (maximum three studies with 30-70 patients per each parameters) , the power of the meta-analysis is somewhat questionable. The main “take home message” of the meta-analysis is that there are insufficient studies and data to draw significant conclusions. Only BMI and cholesterol showed some difference after starting BH4-therapy in meta-analysis, and no other antropometric measurement and nutritional biomarkers benefit was presented. 

Remarks: The manuscript is well written, and the authors are aware of the limitations, that is appropriate interpreted. 

Since there is no meta-analysis data about bone density, mental health, psychosocial functioning and burden of care regarding BH4-treatment, therefore i recommend to delete from the abstract and also from discussion, conclusion sections.

In the introduction section a short paragraph can be added about the charachteristic of patients with BH4-responsive (mild PKU, higher Phe-tolerance, less strict diet).
